# From Fat to Brain: Adiponectin as a Mediator of Neuroplasticity in Depression

**DOI:** 10.3390/biom15121642

**Published:** 2025-11-22

**Authors:** Patrizia Genini, Ilari D’Aprile, Giulia Petrillo, Maria Grazia Di Benedetto, Veronica Begni, Nadia Cattane, Annamaria Cattaneo

**Affiliations:** 1Department of Pharmacological and Biomolecular Sciences, University of Milan, 20133 Milan, Italy; patrizia.genini@unimi.it (P.G.); ilari.daprile@unimi.it (I.D.); giulia.petrillo@unimi.it (G.P.); 2Biological Psychiatry Unit, IRCCS Istituto Centro San Giovanni di Dio Fatebenefratelli, 25125 Brescia, Italy; mdibenedetto@fatebenefratelli.eu (M.G.D.B.); vbegni@fatebenefratelli.eu (V.B.); ncattane@fatebenefratelli.eu (N.C.)

**Keywords:** metabolism, adiponectin, neuroplasticity, depression, physical exercise

## Abstract

Depression is a leading cause of global disability and is increasingly recognized as a multifactorial disorder characterized by fundamental disruptions in neuroplasticity, including diminished hippocampal neurogenesis, impaired synaptic plasticity, and dysregulated stress-response systems. Given the limited efficacy of conventional pharmacological treatments, lifestyle-based interventions—most notably physical exercise—have gained considerable attention for their antidepressant effects, partly mediated by secreted exerkines. Among these, adiponectin has emerged as a particularly compelling candidate linking metabolic regulation to neuroplasticity and mood. Recent evidence suggests that adiponectin contributes to the antidepressant effects of exercise by modulating hippocampal neurogenesis, neuroinflammation, and brain-derived neurotrophic factor (BDNF) signalling. Despite these advances, the mechanisms by which adiponectin influences depression remain incompletely understood. This review synthesizes current knowledge on adiponectin’s role in depression pathophysiology, with emphasis on its capacity to enhance neuroplasticity and hippocampal neurogenesis, and its potential to mediate exercise-induced antidepressant effects via defined molecular pathways. Building on these insights, we discuss adiponectin’s translational promise as both a predictive biomarker of treatment response and a novel therapeutic target. By integrating preclinical and clinical evidence, this review offers a comprehensive perspective on adiponectin’s involvement in depression while identifying critical gaps to guide future mechanistic research.

## 1. Introduction

Depression is one of the leading causes of global disability and arises from a complex interplay of biological and environmental factors that shape its multifaceted aetiology. Stressful life events and chronic stress are among the primary etiological drivers [1], capable of both initiating and sustaining depressive pathology. These factors do not act in isolation: stress and inflammation interact bidirectionally in a synergistic manner that amplifies their pathological impact [2], disrupting key brain functions such as neurogenesis—the generation of new neurons in the adult brain [3,4].

The neurogenic hypothesis of depression posits that impaired hippocampal neurogenesis represents a core mechanism underlying depressive symptomatology and contributes to the therapeutic efficacy of antidepressant treatments [5]. Supporting this view, clinical studies have shown that patients with depression undergoing antidepressant therapy exhibit increased hippocampal neural progenitor cell counts [6,7]. Yet, despite these advances, the precise mechanisms through which stress and inflammation converge to suppress neurogenesis—and how these processes collectively shape the neurobiology of depression—remain incompletely understood.

At the same time, the prevalence of psychiatric disorders, particularly depression, has risen steadily in recent years. This trend can partly be attributed to improved diagnostic accuracy and heightened social awareness, but it also reflects profound changes in lifestyle patterns [8]. Of particular concern is the global shift toward sedentary behaviours, driven by urbanization, technological advances, and changing occupational demands, which have progressively displaced incidental physical activities such as walking, standing, and manual tasks [9,10,11]. Physical inactivity may carry important neurobiological consequences: exercise has been shown to enhance hippocampal neurogenesis [12], lower systemic and central inflammation [13] and improve emotional regulation [14]. Accordingly, decreased daily movement is now considered a modifiable risk factor for depression and related mental health disorders [15]. Consistently, both longitudinal and cross-sectional studies demonstrate that individuals with depression engage in less physical activity and report higher sedentary behaviour compared to healthy controls [16,17,18], with lower physical activity closely linked to core depressive symptoms such as anhedonia, fatigue, and hopelessness, symptoms that directly impair motivation and energy, thereby reinforcing inactivity and creating a vicious cycle of worsening mood and reduced behavioural activation [19,20].

Efforts to understand the biological basis of exercise’s antidepressant effects have identified a wide range of molecular mediators, collectively termed exerkines. Among them, adiponectin has emerged as a particularly intriguing candidate linking physical activity to enhanced neurobiological resilience against depression [21]. Secreted predominantly by adipose tissue and elevated during exercise, adiponectin has been shown to directly influence the central nervous system (CNS) by promoting hippocampal neurogenesis [22] and modulating inflammatory pathways [23]. By intersecting metabolic regulation, brain plasticity, and inflammation—three key processes implicated in depression—this adipokine represents a pivotal mediator of exercise’s neuroprotective and antidepressant effects [24].

Despite increasing interest in adiponectin’s role, significant gaps remain in our mechanistic understanding of how it contributes to exercise-induced resilience against depression. The aim of this review is therefore to synthesize current evidence on adiponectin, clarifying its role as a biological link between physical activity and improved brain health outcomes. Specifically, we highlight its capacity to enhance neuroplasticity and hippocampal neurogenesis, discuss the molecular pathways through which it acts, and explore its potential as a therapeutic target. By integrating findings from preclinical, clinical, and experimental studies, this review provides a comprehensive perspective on adiponectin’s contribution to the neurobiology of depression, with particular emphasis on its mechanistic role in mediating the antidepressant effects of exercise.

## 2. Methods

To identify relevant studies and publications, a literature search was conducted in PubMed (MEDLINE), Google Scholar and Scopus databases, using keywords including neuroplasticity, hippocampal neurogenesis, depression, physical activity, and adiponectin. Publications in English from 2000 to 2025 were considered. Preference was given to peer-reviewed original research articles and meta-analyses addressing molecular and cellular mechanisms linking adiponectin to depression and neuroplasticity.

## 3. Neuroplasticity Dysregulation in Depression

Neuroplasticity represents a fundamental property of the brain, encompassing its ability to adapt both structurally and functionally in response to internal and external stimuli [3]. This multifaceted term includes different processes, from neurogenesis [25] and synaptogenesis [26] to activity-dependent processes like long-term potentiation (LTP) [27], and synaptic remodelling [28]. Collectively, these mechanisms allow the brain to dynamically reshape neuronal circuits and to strengthen or weaken connections in accordance with experience and environmental demands [29,30].

The neuroplasticity hypothesis of depression, which has gained considerable empirical support, posits that impairments in these processes arise from a complex interplay between environmental stressors, such as chronic stress or early-life adversity, and intrinsic factors, including genetic vulnerability, which together compromise the brain’s adaptive capacity [31,32]. The resulting neuroplastic changes occur across multiple biological levels and are context dependent: while in some situations they may promote adaptive responses and resilience, in others—particularly when dysregulated—they contribute to the emergence and persistence of core depressive symptoms, including anhedonia, cognitive impairment, and negative affective states [1].

Structurally, neuroimaging and postmortem studies have consistently revealed alterations in brain regions involved in cognition and emotional regulation, most notably a reduction in hippocampal volume, which is widely regarded as a robust biomarker of depression severity, with the degree of atrophy correlating with the number of depressive episodes and illness chronicity [33]. Interestingly, neuronal loss in the hippocampus has been estimated at approximately 20–35%, while in other regions, such as the prefrontal cortex, volumetric and cellular alterations appear to be even more pronounced [34,35]. These morphological findings may reflect disruptions in adult hippocampal neurogenesis (AHN)—the process by which neural stem cells (NSCs) in the subgranular zone (SGZ) of the dentate gyrus (DG), illustrated in Figure 1, differentiate into mature neurons, thereby supporting hippocampal plasticity and cognitive flexibility [36].

Impairments in AHN limit the brain’s ability to form new connections, adapt to external challenges, and recover from stress-related insults, with mounting evidence suggesting that both stress and depression markedly suppress this process, thereby contributing to the emotional and cognitive dysfunctions observed in patients [37]. For example, rodent studies show that corticosterone administration, a well-established model of depressive-like behaviour, reduces BrdU-positive hippocampal cells in the DG [38], and decreases BDNF levels, thereby impairing AHN through a neuronal autophagy-dependent mechanism, ultimately leading to depressive-like phenotypes [39]. Importantly, AHN is a dynamic and treatment-responsive process: antidepressants, electroconvulsive therapy [40], transcranial magnetic stimulation [41], and exercise [42] have all been shown to enhance hippocampal neurogenesis, underscoring its value as a therapeutic target.

Beyond neurogenesis, synaptic plasticity processes such as LTP and long-term depression (LTD) are also fundamental for hippocampal function, particularly in enabling memory formation, learning, and the integration of newly generated neurons into existing circuits [43] and in supporting the adaptive integration of new neurons into existing circuits [44,45]. LTP, a central mechanism of synaptic strengthening, is triggered by high-frequency stimulation and relies on the coordinated activity of AMPA and NMDA glutamate receptors. While AMPA receptors facilitate sodium influx and depolarization of the postsynaptic membrane, NMDA receptors—upon relief of the magnesium block—permit calcium entry, which in turn activates intracellular signalling cascades involving kinases such as CaMKII and PKC. These cascades promote AMPA receptor phosphorylation and trafficking to the postsynaptic membrane, thereby strengthening synaptic efficacy [46]. Conversely, LTD provides an essential counterbalance by selectively weakening synaptic connections through AMPA receptor internalization, a process triggered by modest NMDA-mediated calcium influx and subsequent activation of protein phosphatases [47]. This bidirectional plasticity maintains network stability, prevents synaptic saturation, and supports adaptive circuit refinement [48].

In depression and chronic stress, however, these mechanisms are significantly disrupted, with evidence from both preclinical and clinical studies pointing to reduced LTP alongside enhanced LTD, particularly within hippocampal circuits [49,50,51]. Such an imbalance contributes to impaired communication between neurons, ultimately manifesting as deficits in memory consolidation, attention, and executive functioning, as well as in motivational processes tied to reward-related circuitry [45,52]. Specifically, reductions in LTP have been linked to the cognitive dysfunction frequently observed in depression [53] while excessive LTD, especially within mesocorticolimbic pathways, may underlie anhedonia and reduced motivation.

The persistence of synaptic strengthening depends not only on early-phase LTP but also on late-phase LTP (L-LTP), which requires the activation of transcriptional and translational pathways such as PKA and MAPK. These cascades enable the synthesis of proteins necessary for structural synaptic modifications, including dendritic spine growth and synaptogenesis, thereby supporting long-term memory consolidation and circuit stabilization [54,55]. Disruptions in these processes impair the maintenance of long-lasting synaptic modifications, leading to reduced brain adaptability and compromised recovery during depression [33], with impairments in L-LTP-related signalling pathways linked not only to cognitive deficits observed in depression but also to the severity of depressive symptoms [53]. A central contributor to these impairments is the disruption of the BDNF—Tropomyosin receptor kinase B (TrkB) signalling pathway: under normal conditions BDNF binds to its receptor TrkB, activating signalling cascades such as MAPK, phospholipase C (PLC) and phosphatidylinositol-3 kinase (PI3K), along with its downstream effector mammalian target of rapamycin (mTOR). This activation directly promotes CREB phosphorylation, which in turn enhances proliferation and differentiation of neural progenitor cells [56].

BDNF signalling is one of the most crucial pathways, not only in promoting synaptic protein translation and dendritic spine growth, but also in modulating neurogenesis [1,57], in particular in the hippocampus [1]. In patients affected by depressive symptoms as well as in the context of exposure to chronic stress, plenty of evidence indicate that BDNF signalling is profoundly dysregulated, which in turn compromise synaptic plasticity and destabilize of neuronal circuits [39,44,58]. Recent meta-analyses and systematic reviews confirm that BDNF levels are significantly lower, both in the blood and in the CNS, in patients with depression compared to controls; moreover, lower peripheral BDNF levels have been associated with more severe symptoms [58,59,60,61,62]. For instance, in the study conducted by Bocchio-Chiavetto et al. [63], patients exhibited serum BDNF concentrations of 29.60 ± 12.41 ng/mL compared to 40.78 ± 11.34 ng/mL in controls. Crucially, rather than being a static vulnerability factor or trait marker, BDNF is considered as a dynamic “state marker” of depression, reflecting current disease activity and treatment responsiveness, as consistently supported by evidence showing that interventions which effectively reduce depressive symptoms often lead to increased BDNF levels [60,61]. For example, in the study by Wolkowitz et al. [64], peripheral serum BDNF concentrations increased from 14.08 ± 5.41 ng/mL at the baseline to 18.75 ± 6.97 ng/mL after eight weeks of antidepressant treatment, reaching the levels observed in controls (20.91 ± 7.07 ng/mL).

Odaira and colleagues (2019) [65] demonstrated that activation of AMP-activated protein kinase (AMPK) promotes hippocampal neurogenesis in a manner comparable to the effects of antidepressants such as selective serotonin reuptake inhibitors (SSRIs). Conversely, loss of AMPK function impairs CREB phosphorylation, resulting in reduced BDNF expression and diminished cell viability, thereby underscoring the pivotal role of this kinase in sustaining neurogenic processes [66].

Emerging evidence suggests that inflammation may exacerbate disruptions in neurotrophic signalling pathways, thereby contributing to the synaptic and structural abnormalities observed in depression. Particularly in the prefrontal cortex and hippocampus, glutamate imbalance is exacerbated by elevated proinflammatory cytokine levels and impaired astrocyte function, which raise excitotoxicity and excessive synaptic pruning [67,68,69,70]. In addition, microglial cells, crucial for their immune role, become activated after chronic stress, releasing pro-inflammatory cytokines [71]. Together, these maladaptive mechanisms reinforce dysfunctional neural circuits and contribute to the emotional and cognitive symptoms of depression by affecting the ability of the brain to adapt to environmental stimuli.

Taken together, these findings underscore that neuroplasticity disruptions—involving neurogenesis, synaptic plasticity, BDNF signalling, energy-sensing pathways, and inflammation—undermine the functional integrity of neural circuits central to cognition, emotional regulation, and stress resilience. Such alterations not only constitute core mechanisms of depression but also correlate with illness severity and chronicity, thereby limiting therapeutic responsiveness and highlighting neuroplasticity as a crucial target for novel interventions [72].

## 4. Adiponectin: From Peripheral Hormone to Brain Modulator

Adiponectin, an adipokine that plays a pivotal role in sustaining systemic metabolic homeostasis, represents one of the most abundant peptide hormones secreted by white adipose tissue (WAT), particularly in response to hormonal cues and nutritional status [73], both of which exert a profound influence on its production and secretion. Lifestyle interventions such as caloric restriction and physical activity have been shown to increase circulating levels of adiponectin by 18–48%, whereas obesity, and especially the accumulation of visceral adiposity, markedly suppresses its expression through a variety of interrelated mechanisms [74]. More specifically, serum adiponectin levels have been reported to be significantly lower in obese individuals compared to normal-weight individuals (7.06 ng/mL vs. 14.57 ng/mL) [75]. Regarding physical exercise as a modulator of adiponectin, a meta-analysis by Yu and colleagues demonstrated that physical activity increased serum adiponectin concentrations by an average of +0.44 µg/mL [76]. With respect to depression, another meta-analysis including six studies and 4220 participants found a significant association between depression and lower adiponectin levels, with a difference of −5.00 µg/mL compared to controls, among European participants [77].

At the cellular level, the secretory capacity of adipocytes depends on their maturation state and is tightly regulated by microenvironmental factors such as hypoxia, nutrient availability, and the prevailing inflammatory milieu [78]. The biological activity of adiponectin is further dictated by its multimeric organization, with high-molecular-weight (HMW) isoforms displaying superior metabolic efficacy, primarily by enhancing insulin sensitivity and conferring cardiovascular protection [79]. Nevertheless, these bioactive isoforms remain particularly vulnerable to disruption under conditions of endoplasmic reticulum (ER) stress and oxidative stress, both of which are accentuated in states of metabolic dysfunction [80,81,82].

Functionally, adiponectin exerts its pleiotropic effects through two distinct receptors (AdipoRs), which, despite their structural similarities, display tissue-specific expression patterns and preferentially activate distinct signalling pathways. AdipoR1 primarily engages AMPK to regulate energy homeostasis and mitochondrial function, whereas AdipoR2 predominantly signals through peroxisome proliferator-activated receptor alpha (PPARα), modulating lipid metabolism and exerting anti-inflammatory effects [83]. These complementary pathways converge to produce insulin-sensitizing and anti-inflammatory effects in key metabolic organs: in the liver, adiponectin suppresses gluconeogenesis while promoting fatty acid oxidation [84,85], whereas in skeletal muscle, it enhances glucose uptake and β-oxidation [86]. Notably, these metabolic benefits are further potentiated by adiponectin’s immunomodulatory capacity, which attenuates the production of pro-inflammatory cytokines while simultaneously enhancing anti-inflammatory mediators such as IL-10 [87,88,89,90]. From a pharmacological perspective, the adiponectin pathway demonstrates substantial therapeutic potential, as exemplified by thiazolidinediones (TZDs), PPARγ agonists widely used in the management of type 2 diabetes. TZDs increase circulating adiponectin levels by promoting both its transcription and secretion, thereby improving insulin sensitivity and alleviating metabolic dysfunction [91,92].

Endocrine regulation further modulates adiponectin dynamics, with oestrogens shown to upregulate its expression, which may partly explain the higher plasma concentrations typically observed in females [93]. Conversely, chronic low-grade inflammation, characterized by elevated pro-inflammatory cytokines such as tumour necrosis factor alpha (TNF-α) and interleukin-6 (IL-6), suppresses adiponectin gene expression via activation of nuclear factor kB (NF-κB), thereby exacerbating metabolic dysfunction and promoting insulin resistance [88,89].

Beyond its well-established role in metabolic regulation, adiponectin has also emerged as a critical player in neuropsychiatric disorders. Unlike many peripheral metabolic hormones, adiponectin can cross the blood–brain barrier (BBB) and exert direct effects on CNS, particularly within the hippocampus [94,95], a capability confirmed by its detection in the cerebrospinal fluid (CSF) of both rodents [38] and humans [96], although its concentration in human CSF is approximately 1000-fold lower than in serum [97]. These findings support the notion that adiponectin functions as a key molecular messenger, linking peripheral metabolic cues to central neurobiological adaptations. In the brain, AdipoRs are widely expressed in regions critical for mood regulation, neuroplasticity, and energy homeostasis, including the hippocampus, cerebral cortex, hypothalamus, pituitary gland, and brainstem [98]. Rodent studies examining hippocampal synaptosomes indicate that AdipoR1 is present in both presynaptic and postsynaptic compartments, whereas AdipoR2 is predominantly presynaptic, suggesting potential roles for these receptors in synaptic function, learning, and memory [99]. Moreover, AdipoRs are also expressed in glial cells such as astrocytes and microglia, indicating that adiponectin may influence neuroinflammatory and immune mechanisms relevant to depression [98].

Clinical evidence demonstrates an inverse relationship between circulating adiponectin levels and depression, with lower concentrations associated not only with disease presence but also with greater symptom severity, suggesting a potential causal role for this adipokine in depressive pathophysiology [100,101,102,103]. This association is further supported by longitudinal studies in patients treated with interferon alpha (IFN-α) for hepatitis C virus (HCV), in which the emergence of depressive symptoms during treatment correlated with progressively rising adiponectin levels, likely representing a compensatory response to treatment-induced inflammation, mediated by the hormone’s anti-inflammatory properties [104].

Preclinical studies provide mechanistic insight into adiponectin’s neurobehavioral effects, demonstrating its capacity to modulate anxiety-like behaviours in specific brain regions [105]. Disruption of adiponectin signalling in the CNS has been associated with neurobehavioral dysfunctions. For example, our recent study [106] showed that prenatal stress reduced AdipoR1 and AdipoR2 expression in both liver and ventral hippocampus, coinciding with social deficits and elevated inflammation. Importantly, adiponectin exerts potent anti-inflammatory effects within the CNS by downregulating microglial activation and suppressing pro-inflammatory cytokines, including TNF-α and IL-6, thereby creating a more favourable environment that facilitates synaptic plasticity and cognitive performance [107,108,109].

The expression of AdipoRs in the brain modulates the function of several neurotransmitters and neuropeptides. Findings show that AdipoR1 is present in serotonin neurons in the dorsal raphe nucleus, where deletion of AdipoR1 in these area leads to reduced expression of the serotonin synthesis enzyme Tryptophan Hydroxylase 2 (TPH2), decreased serotonin levels and altered Serotonin Transporter (SERT) expression, resulting in impaired serotonin transmission and depression-like behaviours [110]. Similarly, dopamine neurons in the ventral tegmental area express AdipoR1 and loss of AdipoR1 pathway activity has been shown to increase dopamine activity and anxiety-like behaviours [99,105]. Regarding glutamatergic transmission, recent evidence shows that adiponectin knockout mice display cognitive deficits and impaired synaptic plasticity in the hippocampus, accompanied by altered levels of presynaptic and postsynaptic proteins involved in glutamatergic neurotransmission. These deficits can be rescued by adiponectin receptor agonists, indicating a regulatory role for adiponectin in the glutamate receptor expression and synaptic function [99,111].

Although adiponectin is generally referred to an anti-inflammatory and neuroprotective molecule, several clinical studies have described the so-called “adiponectin paradox”, whereby higher circulating adiponectin levels are paradoxically associated with cognitive decline, Alzheimer’s disease and frailty in older adults. In more details, different studies have reported that elevated adiponectin levels predict incident Alzheimer’s disease, especially in women and in individuals with amyloid pathology or low body mass index (BMI) [112,113,114]. However, other studies have found that lower adiponectin levels are associated with worse cognitive performance, suggesting a possible protective role of adiponectin in brain functions [115,116]. This opposite association highlights the complexity of adiponectin signalling in neurodegenerative processes and underscores the need for further studies.

In addition to its link with depressive and anxiety symptoms, adiponectin may also serve as a predictor of pharmacological treatment response [117]. Low baseline plasma adiponectin levels have been associated with a rapid clinical response to ketamine, with lower initial concentrations correlating with greater treatment efficacy [118]. Similarly, in treatment-resistant depression, clinical improvements following interventions such as electroconvulsive therapy are accompanied by increases in adiponectin levels. Moreover, baseline adiponectin concentrations have been shown to correlate with BDNF levels, suggesting that both biomarkers may dynamically reflect depressive state. Collectively, these findings indicate that adiponectin and BDNF fluctuate together in relation to symptom severity, highlighting the central role of neuroplasticity in depression [119], as will be further discussed in the following section.

## 5. Linking Exercise, Adiponectin, and Neuroplasticity

### 5.1. How Exercise Enhances Adiponectin Levels: Cellular and Molecular Mechanisms

When the body undergoes physical exercise, a cascade of cellular and molecular changes is initiated, establishing exercise as one of the most effective non-pharmacological strategies for promoting brain health [21]. Among the myriad bioactive compounds released in response to physical activity, collectively termed exerkines [120], adiponectin has garnered particular attention due to its central role at the intersection of metabolic regulation, neuroplasticity, and mood modulation [121].

Physical exercise modulates adiponectin production in a manner strongly dependent on its type, intensity and duration. Aerobic and endurance training, particularly when performed regularly over 3 months consistently elevates circulating adiponectin levels [122], reflecting improved metabolic and anti-inflammatory adaptations. Acute, high-intensity exercise can trigger transient increases in adiponectin, especially in young and fit individuals [123,124,125,126,127,128], although findings remain inconsistent across studies, with some reporting modest rises and others no significant changes or even slight reductions [124,129,130]. In contrast, chronic moderate-intensity training produces more stable and pronounced increases in adiponectin, particularly in individuals with obesity or type 2 diabetes [131,132,133]. Light-intensity physical activity may also enhance adiponectin levels, although responses tend to be more variable in mixed populations [128]. Overall, aerobic modalities appear more effective than combined training in raising adiponectin concentrations, though the magnitude of change often depends on the duration of the intervention, the baseline metabolic status and other individual characteristics [121,131,134,135].

Combined training can also lead to increases in adiponectin, particularly when paired with aerobic exercise. However, these effects are generally less consistent and smaller compared to aerobic training alone [127,134,136,137].

It is important to note that differences in adiponectin production depend not only on lifestyle-related factors but also on an individual’s genetic background, particularly on genetic variants within the ADIPOQ gene, which can influence adiponectin expression, circulating levels and biological activity. Several studies across diverse populations have investigated the impact of ADIPOQ polymorphisms on variations in adiponectin concentrations, insulin sensitivity, diabetes susceptibility and, importantly, on the physiological response to physical exercise. Notably, multiple ADIPOQ variants, including rs2241766 [138], rs1501299 [138,139,140], rs16861205 [141], rs266729 [139,142] and rs17300539 [143,144] have been associated with longitudinal changes in serum adiponectin levels, BMI, fat mass and lipid profile following exercise interventions. Importantly, specific allele carriers tend to exhibit greater reductions in body fat and more pronounced metabolic improvements after aerobic training compared to others, for example, C allele carriers of rs266729 [142] and homozygous A allele carriers of rs17300539 [143,144]. These findings emphasize the role of ADIPOQ genetic variability in modulating exercise responsiveness and can partly explain the substantial interindividual variability observed in exercise-induced metabolic outcomes.

Interestingly, adiponectin behaves differently from many other adipokines, with its circulating levels rising in response to physical activity despite being inversely correlated with body fat [121]. While part of this exercise-induced increase may be mediated by weight loss, elevated adiponectin levels have been observed following physical activity in healthy individuals, people with obesity, and in various rodent models, suggesting that exercise can modulate adiponectin levels independently of changes in adiposity [145].

Exercise modulates multiple molecular pathways that operate both at the metabolic and central nervous system levels. It triggers a cascade of signals that collectively regulate the balance between stress- and anti-stress mediators. This includes increased circulating levels of BDNF [21], irisin [146] and β-endorphins [147], together with decreased concentrations of pro-inflammatory cytokines and cortisol [148], which contribute to an overall homeostatic shift toward resilience leading to an improved mood regulation. In this context, adiponectin acts synergistically within this complex network by enhancing insulin sensitivity, reducing neuroinflammation and supporting BDNF signalling. Along with other exercise-induced mediators, such as irisin and insulin [149], adiponectin helps to maintain metabolic balance and promote neuronal health. Together, all these molecules enhance brain energy metabolism, increase neurotrophic support and foster resilience to stress [150]. Therefore, the antidepressant effects of exercise are more likely to arise from the integrated action of multiple mediators, with adiponectin representing one of the key components linking peripheral metabolic adaptations to central mechanisms of neuroplasticity.

The precise mechanisms through which exercise enhances adiponectin secretion remain incompletely understood, although several hypotheses have emerged from both rodent and human studies [107,151,152]. Central to these mechanisms is adipose tissue, comprising brown adipose tissue (BAT), which is limited in distribution and primarily involved in thermogenesis [153], and WAT, which serves both as an energy reservoir and as an active endocrine organ.

As illustrated in Figure 2, during exercise, increased energy demand triggers lipolysis in WAT, mobilizing stored triacylglycerols to provide fatty acids to working muscles and other tissues [154]. This activation of WAT is accompanied by functional adaptations, including enhanced mitochondrial activity and increased glucose uptake through the translocation of glucose transporter type 4 (GLUT4) [155], which together contribute to improved systemic insulin sensitivity and energy homeostasis [156]. Consequently, circulating insulin levels decrease, fostering a more insulin-sensitive and less inflammatory microenvironment within adipose tissue, thereby favouring the expression of the ADIPOQ gene that encodes adiponectin [157,158]. Moreover, insulin itself directly stimulates ADIPOQ expression, particularly in insulin-sensitive individuals, as demonstrated by increased ADIPOQ mRNA following acute insulin administration in subcutaneous adipose tissue [159]. Thus, exercise-induced improvements in insulin sensitivity substantially contribute to adiponectin upregulation.

Within adipocytes, ADIPOQ transcription is regulated by pathways involving PPARγ and AMPK, both of which are activated by physical activity [160,161,162] and by reductions in pro-inflammatory signalling [163]. PPARγ, a nuclear receptor essential for adipocyte differentiation and lipid metabolism, acts as a principal transcriptional regulator of ADIPOQ [164]. Upon binding free fatty acids released by lipolysis [165], PPARγ heterodimerizes with the retinoid X receptor (RXR) and binds to PPAR response elements (PPREs) in the ADIPOQ promoter, robustly enhancing its transcription [166]. Pharmacological activation of PPARγ has been shown to increase circulating adiponectin, particularly the metabolically protective HMW isoform, in both mice and humans, through both direct upregulation of ADIPOQ transcription and enhanced expression of molecular chaperones necessary for adiponectin multimerization and secretion [164,167,168]. Animal studies further corroborate this link, demonstrating that PPARγ upregulation following weight loss correlates with both increased adiponectin and improved insulin sensitivity [169].

Simultaneously, AMPK, a key cellular energy sensor, is activated during exercise in response to an increased AMP/ATP ratio driven by heightened energy demand and rapid ATP consumption required for muscle contraction and other cellular activities [170]. Once activated, AMPK enhances PPARγ activity and may also upregulate the expression of PPARG, the gene encoding PPARγ [171], thereby further stimulating adiponectin production. This crosstalk between AMPK and PPARγ establishes a mechanistic link between exercise and adiponectin upregulation [107].

Collectively, these coordinated pathways promote ADIPOQ transcription, secretion, and multimerization, contributing to improved neuroplasticity, enhanced metabolic adaptation, and alleviation of depressive symptoms [172,173]. Notably, adiponectin itself can activate AMPK within the CNS, leading to further beneficial effects on neuronal function, which will be discussed in the following section.

### 5.2. Molecular Mechanisms Underlying Adiponectin’s Effects on Neurogenesis and Neuroplasticity

As previously discussed, numerous studies have identified adiponectin as a key molecular mediator linking physical exercise to neurogenesis and mood regulation [38,95]. For instance, in a diabetic mouse model, voluntary running failed to restore hippocampal neurogenesis in the absence of adiponectin [12], while reduced serum adiponectin levels have been associated with impaired hippocampal neurogenesis [174], highlighting its essential role in mediating exercise-induced neurogenic effects and its broader relevance beyond metabolic regulation. The involvement of adiponectin signalling is further supported by evidence that pharmacological activation via AdipoRon, a selective AdipoRs agonist, alleviates depression- and anxiety-like behaviours and enhances hippocampal plasticity [26,175]. Notably, in vivo studies demonstrate that voluntary running increases hippocampal cell proliferation and neurogenesis only in wild-type mice, whereas adiponectin knockout mice fail to exhibit these antidepressant-like effects. Similarly, in vitro exposure of NSCs to AdipoRon for 48 h activates key signalling pathways involved in neurogenesis and synaptic plasticity [176], confirming that adiponectin is necessary for both mood-enhancing and neuroplastic benefits [95].

Collectively, these findings position adiponectin as a central player in translating physical activity into cognitive and emotional resilience through mechanisms that converge on hippocampal neurogenesis and plasticity. The question then arises: through which specific molecular pathways are these effects mediated? Mechanistically, adiponectin engages multiple signalling cascades that converge on hippocampal neurogenesis and synaptic plasticity, as illustrated in Figure 3. Upon activation of AdipoR1 and AdipoR2, downstream pathways include (i) metabolic sensors such as the AMPK cascade and its downstream kinases, (ii) survival and anti-apoptotic signalling, (iii) developmental and stem cell fate regulators such as Notch, (iv) neurotrophic factor pathways including STAT3/BDNF, (v) anti-inflammatory signalling via suppression of NF-κB and (vi) antioxidative and mitochondrial-protective pathways. Together, these interconnected networks translate metabolic and exercise-related signals into structural and functional adaptations in the hippocampus.

Among these pathways, the AdipoR1-AMPK axis is the most extensively characterized, functioning as a central metabolic sensor linking energy status to neuronal adaptation. Exercise promotes hippocampal plasticity, protects neurons from stress-induced atrophy, enhances dendritic remodelling, and alleviates depressive-like behaviours in wild-type mice via AMPK phosphorylation. In contrast, adiponectin knockout mice fail to activate this pathway, underscoring the critical role of AdipoR1-AMPK signalling in mediating exercise-induced neuronal growth and energy metabolism [24].

Further insights into AMPK’s role in NSC regulation come from in vitro studies. B. Liu et al. demonstrated that AdipoRon treatment significantly increases phosphorylated AMPK expression in NSCs, whereas co-administration of Compound C, a selective AMPK inhibitor, abolishes this effect. AdipoRon also activates the transcription factor CREB, whose phosphorylation is suppressed by Compound C, indicating that adiponectin promotes NSC proliferation via the AMPK/CREB axis [176]. Additional evidence highlights the involvement of AMPK downstream kinases in sustaining neurogenic actions: Whitaker and Cook (2021) [177] reported that adiponectin stimulates adult human NSC proliferation via AMPK-mediated activation of p38MAPK, a kinase with both proliferative and anti-proliferative roles. Inhibition of p38MAPK significantly attenuates adiponectin-induced proliferation, without affecting basal cell growth, while phosphorylation of GSK-3β induced by adiponectin is dependent on p38MAPK activity, enhancing transcription of neurogenesis-related genes [178]. These findings reinforce the AdipoR1-AMPK axis as a critical metabolic translator, converting physical effort into neurogenic benefits.

Adiponectin also promotes NSC survival and anti-apoptotic signalling. Under stressful conditions, such as high glucose exposure, adiponectin downregulates apoptotic regulators including p21, p53, and c-Myc, while upregulating neurogenic factors, thereby shifting the balance from cell death toward regeneration [179]. Blockade of AdipoR1 abolishes this protective effect, resulting in increased apoptosis, confirming that receptor engagement is essential for maintaining NSC viability under metabolic stress.

In addition to survival signalling, adiponectin modulates developmental regulators of stem cell fate, with the Notch pathway emerging as a key mediator [180]. Early studies demonstrated that a homolog of adiponectin upregulates ADAM10 and ADAM17, metalloproteinases essential for Notch receptor activation, in the hippocampus [181]. Subsequent research showed that physical exercise increases expression of Notch1/2 receptors, ADAM10, and downstream targets in wild-type mice, whereas this effect is absent in adiponectin knockout mice, indicating that adiponectin is necessary for exercise-induced Notch activation [175]. This activation is further regulated via PPARα signalling, emphasizing the role of the adiponectin-Notch axis in mediating neurogenesis, cognitive function, and the protective effects of exercise against chronic stress-induced impairments in hippocampal plasticity.

Beyond neurogenesis and survival, adiponectin influences broader neuroplasticity processes, modulating synaptic structure and hippocampal circuit remodelling. Direct brain infusion of adiponectin increases dendritic spine density and arborization in hippocampal granule neurons, key indicators of synaptic efficacy [22]. Bloemer et al. (2019) [99] demonstrated that AdipoRs signalling in hippocampal synapses is reduced in adiponectin knockout mice, resulting in deficits in memory and cognitive tasks. In these mice, reductions in LTP correlate with decreased glutamatergic receptor subunits in the hippocampus, supporting the role of adiponectin in synaptic strengthening. Similarly, Pousti et al. (2018) [108] showed that adiponectin enhances presynaptic release probability and induces chemical LTP in the dentate gyrus, further confirming its role in synaptic plasticity.

Additional pathways contribute to adiponectin’s neuroprotective effects, most notably the BDNF pathway, which is critical for neuronal survival, synaptic plasticity, and hippocampal function [182]. While exercise independently enhances BDNF signalling [11], adiponectin may facilitate this effect via STAT3-mediated upregulation of BDNF in astrocytes. Inhibition of either STAT3 or BDNF abolishes the neural and behavioural benefits, highlighting the essential role of the adiponectin-STAT3-BDNF axis in mediating neuroplasticity [183]. In corticosterone-treated depressive-like mice, AdipoRon restores hippocampal expression of neurotrophic genes, including BDNF, VEGFα, IGF1, and NGF, while activating AMPK signalling, emphasizing its targeted role in restoring stress-induced impairments rather than broadly enhancing growth [184].

Inflammation represents another critical dimension in adiponectin-mediated neuroplasticity. Physical exercise reverses stress-induced changes in hippocampal microglia and neuroinflammation, exerting antidepressant effects [71]. Voluntary running promotes an anti-inflammatory microglial phenotype, suppressing neuroinflammation in the hippocampus, and these effects are dependent on the adiponectin/AdipoR1 pathway [107].

According to Chabry et al. (2015) [185], adiponectin levels increase following voluntary exercise, promoting a microglial anti-inflammatory state that reduces neuroinflammation as well as anxiety- and depressive-like behaviours. A subsequent study from the same group further demonstrated that this effect involves AMPK phosphorylation, which is downregulated by pro-inflammatory triggers such as lipopolysaccharide (LPS) and upregulated by adiponectin in microglial cells. Moreover, adiponectin activates AMPK and suppresses NF-κB signalling, thereby reducing the synthesis of pro-inflammatory cytokines and chemokines [23].

Adiponectin exerts potent antioxidant and mitochondrial-protective effects that are fundamental for neuroprotection. Kadowaki and colleagues demonstrated that oxidative stress is markedly elevated in AdipoR1- and AdipoR2-deficient mice, providing compelling evidence that the adiponectin–AdipoRs signalling pathway plays a pivotal role in counteracting oxidative damage and maintaining redox balance [186]. Experimental evidence indicates that adiponectin reduces the generation of reactive oxygen species (ROS) primarily by enhancing the activity of key antioxidant enzymes such as superoxide dismutase (SOD), catalase and glutathione peroxidase, thereby strengthening endogenous antioxidant defences [187]. Beyond its antioxidant functions, adiponectin also regulates mitochondrial homeostasis and quality control, mechanisms essential for sustaining neuronal energy metabolism. Activation of the AdipoR1-PGC-1α signalling axis has been shown to prevent mitochondrial oxidative damage, restore mitochondrial biogenesis and promote the removal of damaged mitochondria via mitophagy, thus limiting ROS accumulation and preserving mitochondrial efficiency [188,189,190,191]. Moreover, the suppression of NF-κB signalling due to the anti-inflammatory effects of adiponectin further reduces ROS generation and prevents neuronal cytotoxicity [192]. Through these combined actions, adiponectin lowers oxidative stress, preserves mitochondrial function, mitigates cellular injury and contributes to neuronal survival and synaptic plasticity within the hippocampus.

To complement these findings, pharmacological inhibition through specific AdipoRs antagonists has been employed to dissect the contribution of adiponectin signalling to hippocampal function. This bidirectional approach—demonstrating both the beneficial effects of receptor activation and the detrimental consequences of its blockade—provides strong evidence for a causal role of adiponectin in mediating the neuroplastic and cognitive benefits of exercise. For example, administration of the AdipoRs antagonist ADP400 to mice undergoing physical training reduces synaptic protein expression and abolishes exercise-induced memory improvements, while anxiety-related behaviors remain unaffected [26]. Independent studies further support the essential role of adiponectin signalling: peripheral inhibition of AdipoRs via ADP400 leads to a broad spectrum of deleterious effects, including increased β-amyloid accumulation in the hippocampus in Alzheimer’s disease models, heightened anxiety-like behaviour, impaired cognitive performance, and disrupted hippocampal neurogenesis [193].

Taken together, these findings highlight that beyond its established role in metabolic regulation, adiponectin is a central mediator of neuroplastic processes. By translating external beneficial stimuli, such as physical activity, into enhanced hippocampal plasticity, resilience to stress, and improved cognitive and emotional function, adiponectin orchestrates multiple intersecting signalling pathways—including AMPK, BDNF, and Notch. These pathways do not act in isolation; rather, they dynamically interact to stimulate neurogenesis while safeguarding neuronal integrity against stress and inflammatory damage. This sophisticated profile positions adiponectin as a promising molecular bridge between lifestyle factors and brain health, with potential therapeutic relevance for neuropsychiatric disorders.

## 6. Conclusions and Future Directions

In this review, we have highlighted adiponectin as a pivotal hormone at the crossroads between metabolic health, brain plasticity, and emotional regulation—domains that are profoundly disrupted in depression. Both preclinical and clinical evidence indicates that, upon crossing the BBB, adiponectin can modulate synaptic plasticity, neurogenesis, and neuronal survival, processes central to the pathophysiology of depressive disorders. These neuroprotective effects are mediated through multiple pathways, most notably AMPK, PPAR-α, and BDNF signalling, alongside Notch activation and the upregulation of anti-apoptotic markers.

The rationale of this review was to examine the mechanisms by which exercise-induced adiponectin exerts its beneficial effects on the brain, particularly in counteracting the neural dysfunctions associated with depression. What emerges is a molecule with considerable translational promise, meriting deeper exploration.

First, its potential as a biomarker for both depression severity and treatment response has gained increasing empirical support. For example, it is known that systemic adiponectin abnormalities predicted a favourable response to ketamine in patients with major depressive disorder, a finding echoed in studies of SSRI- and SNRI-treated individuals. In the context of rising treatment-resistant depression and the modest efficacy of conventional antidepressants, interest is growing in alternative or complementary strategies that target the biological underpinnings of mood disorders. Because circulating adiponectin is responsive to lifestyle modifications, it could serve as a useful stratification tool to identify patients most likely to benefit from interventions such as structured exercise. Physical activity not only elevates adiponectin levels but also enhances mood and neuroplasticity, raising the prospect of adiponectin as a dynamic biomarker to monitor efficacy in exercise-based treatments. Similarly, pharmacological approaches that enhance adiponectin signalling, such as AdipoRon, a synthetic AdipoR agonist, offer the possibility of mimicking the neurogenic and mood-regulating benefits of exercise in patients unable to engage in regular physical activity. Dietary strategies, including adherence to the Mediterranean diet, have also been linked to higher adiponectin levels and lower depressive symptomatology, further supporting its role as a potential biomarker of response. Monitoring adiponectin in these contexts may be particularly valuable in depression subtypes with metabolic comorbidities, such as insulin resistance or obesity, where identifying a distinct “metabotype” could guide more personalized and integrative therapeutic approaches.

Second, while preclinical data strongly support the neuroprotective and pro-neurogenic actions of adiponectin, substantial translational research is still required to determine how best to harness these pathways clinically. Future mechanistic studies should focus on AdipoR function in brain regions central to mood regulation, such as the hippocampus and prefrontal cortex and explore how adiponectin signalling intersects with other neurotrophic and inflammatory networks to identify new drug targets. Clinically, randomized trials are needed to test whether interventions that elevate adiponectin—through exercise, diet, or pharmacological AdipoR agonists can improve depressive symptoms.

Given the higher baseline adiponectin levels typically observed in women and its possible role in stress resilience, it will also be important to examine sex-specific responses and whether adiponectin modulation benefits particular subgroups, such as patients with metabolic syndrome or atypical depression.

Metformin, an insulin sensitizer widely used in type 2 diabetes, provides a useful pharmacological model in this regard. Beyond its metabolic effects, Metformin increases adiponectin levels and exerts neuroprotective actions via AMPK activation, with evidence of reversing depressive symptoms and normalizing Hypothalamic–Pituitary–Adrenal Axis dysfunction. However, its limited selectivity for AdipoRs underscores the need for more targeted strategies. Nonetheless, Metformin illustrates the therapeutic potential of leveraging adiponectin pathways in developing next-generation treatments for depression and its metabolic comorbidities.

Looking forward, interdisciplinary collaboration will be crucial for translating adiponectin research into clinical practice. Integrating psychiatry, endocrinology, and metabolism could yield innovative interventions that address both the mental and physical dimensions of depression. Advances in biomarker science, particularly the use of adiponectin in combination with other metabolic and inflammatory indicators, may further refine depression subtyping and support more precise, personalized treatment strategies. Ultimately, by bridging the gap between metabolic dysfunction and mood disorders, adiponectin may pave the way toward a new generation of “metabocentric” therapies that expand our approach to treating depression and related conditions.

## Figures and Tables

**Figure 1 biomolecules-15-01642-f001:**
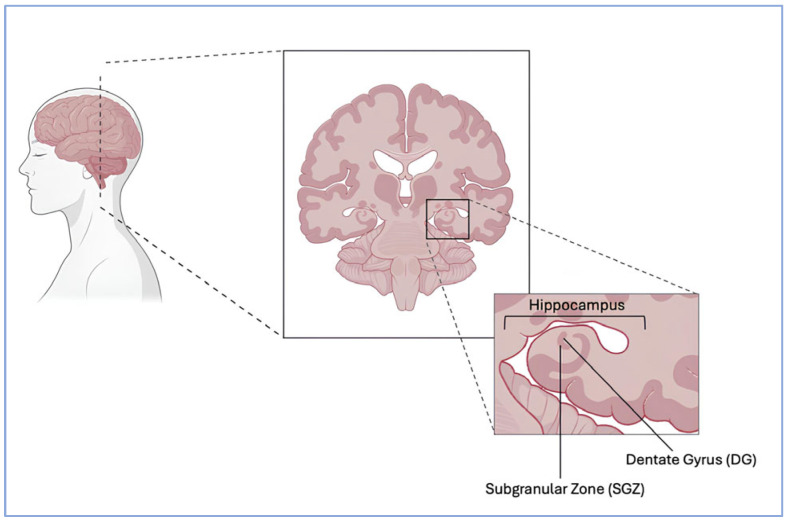
Localization of the dentate gyrus (DG) and subgranular zone (SGZ) within the hippocampus.

**Figure 2 biomolecules-15-01642-f002:**
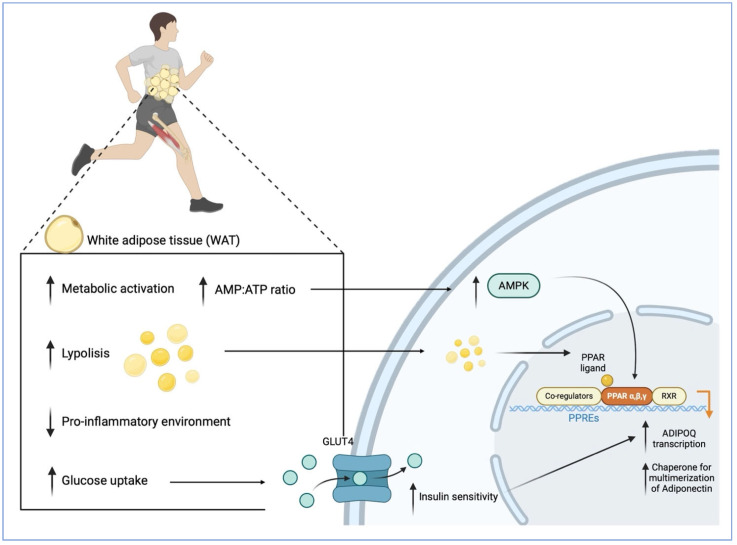
Adiponectin production induced by physical exercise in WAT.

**Figure 3 biomolecules-15-01642-f003:**
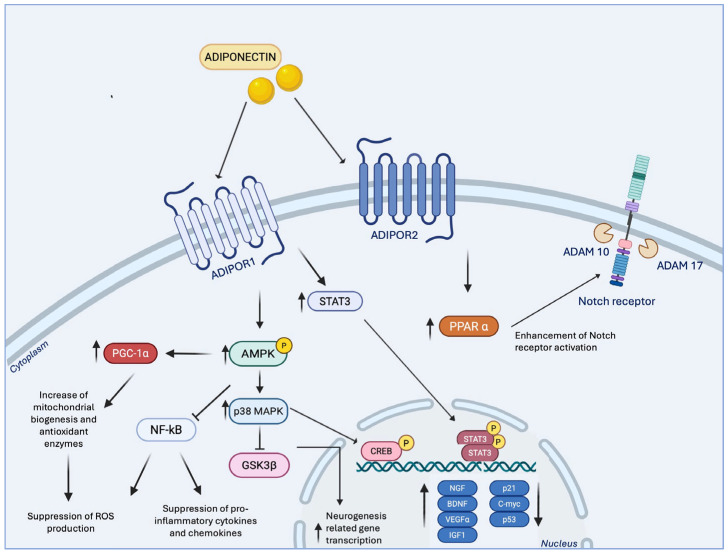
Adiponectin Signalling Pathways Involved in Neurogenesis and Neuroplasticity.

## Data Availability

No new data were created or analyzed in this study. Data sharing is not applicable to this article.

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
