# Peer review of "From Fat to Brain: Adiponectin as a Mediator of Neuroplasticity in Depression"

_biomolecules, 2025, doi:10.3390/biom15121642_

Round 1
Reviewer 1 Report
Comments and Suggestions for Authors
The manuscript entitled “From Fat to Brain: Adiponectin as a Mediator of Neuroplasticity in Depression” proposes a review of the literature data concerning the roles of adiponectin in depression pathophysiology, with emphasis on the capacity of adiponectin to enhance neuroplasticity and neurogenesis. A synthesis of current knowledges on the potential of adiponectin to mediate exercise-induced antidepressant effects via defined molecular pathways is developed.
The content of the manuscript is well described and provides relevant original data to show how adiponectin has emerged as a particularly compelling candidate that may link metabolic regulations to neuroplasticity. By integrating preclinical and clinical evidence, this review offers a comprehensive perspective on adiponectin protective effects in depression and adiponectin roles as both a predictive biomarker and a novel therapeutic target.
The following revisions are proposed in order to improve the content of the manuscript:
Major revisions:
1/ Lines 110-116: A figure presenting the different anatomical structures of the brain may help to visualize that the Authors describe in the text.
2/ Lines 174-181: The Authors need to indicate quantitative data regarding the differences in circulating levels of BDNF in healthy individuals vs patients with depression. Indeed, it is mentioned at different parts of the text that low circulating levels of BDNF have been associated with more severe symptoms of depression and that BDNF may serve as a “dynamic state marker of depression”. However, there a lack of quantitative blood levels expected in both physiological and pathological conditions.
3/ Lines 210-211 and in Chapter 4 linking Adiponectin, Exercise and Depression: The Authors also need to indicate quantitative data regarding the differences in circulating levels of Adiponectin in healthy individuals vs patients with depression, and in conditions without or with exercise. As mentioned above for BDNF, if Adiponectin may serve as a circulating biomarker, it is important to mention which circulating concentrations are considered normal/physiological and those considered low/pathological.
4/ Chapter 4 linking Adiponectin, Exercise and Depression (lines 308-313): Although the manuscript is dedicated to Adiponectin, the Authors need to discuss that the beneficial effects of exercise may also be mediated through other mediators than Adiponectin. Adiponectin contributes to a global effect of exercise that decreases several other mediators of stress. It is a question of homeostasis between levels of indicators of depression and anti-depression signals. This point needs to be discussed to better consider the place/roles of Adiponectin among different signals regulated by exercise and related to the control of stress.
5/ Chapter 4 linking Adiponectin, Exercise and Depression (Figure 2 and the associated text): It would be relevant to complete the text and the Figure 2 with the description of some molecular targets involved in oxidative stress regulation, given that Adiponectin is also known to exert antioxidant activities. Indeed, oxidative stress and inflammation are known to act bidirectionally (as mentioned Line 40) and amplify cellular dysfunctions during depression. Thus the dual anti-inflammatory and antioxidant properties of Adiponectin may be involved.
Minor Revisions:
6/ Lines 9016-917. The reference 128 needs to be formatted to delete “(Noisy-le-grand)” indication.
Author Response
The manuscript entitled “From Fat to Brain: Adiponectin as a Mediator of Neuroplasticity in Depression” proposes a review of the literature data concerning the roles of adiponectin in depression pathophysiology, with emphasis on the capacity of adiponectin to enhance neuroplasticity and neurogenesis. A synthesis of current knowledges on the potential of adiponectin to mediate exercise-induced antidepressant effects via defined molecular pathways is developed.
The content of the manuscript is well described and provides relevant original data to show how adiponectin has emerged as a particularly compelling candidate that may link metabolic regulations to neuroplasticity. By integrating preclinical and clinical evidence, this review offers a comprehensive perspective on adiponectin protective effects in depression and adiponectin roles as both a predictive biomarker and a novel therapeutic target.
We thank the reviewer for appreciating our work.
The following revisions are proposed in order to improve the content of the manuscript:
Major revisions:
1/ Lines 110-116: A figure presenting the different anatomical structures of the brain may help to visualize that the Authors describe in the text.
Response 1. We thank the reviewer for this suggestion. The suggested figure has now been added on page 5.
2/ Lines 174-181: The Authors need to indicate quantitative data regarding the differences in circulating levels of BDNF in healthy individuals vs patients with depression. Indeed, it is mentioned at different parts of the text that low circulating levels of BDNF have been associated with more severe symptoms of depression and that BDNF may serve as a “dynamic state marker of depression”. However, there a lack of quantitative blood levels expected in both physiological and pathological conditions.
Response 2. We thank the reviewer for the valuable suggestion. We agree that providing quantitative data on the differences in circulating BDNF levels between individuals and patients with depression is important to improve the clarity of our manuscript. However, it should be noted that the literature shows considerable variability in reported values due to several methodological issues and biological factors, such as for example the type of biological matrix analysed (serum vs plasma), sample handling and assay techniques. Nevertheless, indicative quantitative data on BDNF concentrations in serum have been reported, and we have modified this section accordingly (page 4-5, line 184-193) to include representative values observed in depressive and control conditions. Now, the paragraph reads: “For instance, in the study conducted by Bocchio-Chiavetto et al. (Bocchio-Chiavetto et al., 2010), patients exhibited serum BDNF concentrations of 29.60 ± 12.41 ng/mL compared to 40.78 ± 11.34 ng/mL in controls. Crucially, rather than being a static vulnerability factor or trait marker, BDNF is considered as a dynamic "state marker" of depression, reflecting current disease activity and treatment responsiveness, as consistently supported by evidence showing that interventions which effectively reduce depressive symptoms often lead to increased BDNF levels (Correia et al., 2023; Kishi et al., 2018). For example, in the study by Wolkowitz et al. (Wolkowitz et al., 2011), peripheral serum BDNF concentrations increased from 14.08 ± 5.41 ng/mL at the baseline to 18.75 ± 6.97 ng/mL after eight weeks of antidepressant treatment, reaching the levels observed in controls (20.91 ± 7.07 ng/mL).”
3/ Lines 210-211 and in Chapter 4 linking Adiponectin, Exercise and Depression: The Authors also need to indicate quantitative data regarding the differences in circulating levels of Adiponectin in healthy individuals vs patients with depression, and in conditions without or with exercise. As mentioned above for BDNF, if Adiponectin may serve as a circulating biomarker, it is important to mention which circulating concentrations are considered normal/physiological and those considered low/pathological.
Response 3. We thank the reviewer for this insightful comment. Accordingly, we have now added quantitative data regarding circulating adiponectin levels in obese individuals, patients with depression and in response to exercise. These data have been incorporated into the revised manuscript (page 6, lines 227-234) and the section now reads: “More specifically, serum adiponectin levels have been reported to be significantly lower in obese individuals compared to normal-weight individuals (7.06 ng/mL vs. 14.57 ng/mL) (Jonas et al., 2017). Regarding physical exercise as a modulator of adiponectin, a meta-analysis by Yu and colleagues demonstrated that physical activity increased serum adiponectin concentrations by an average of +0.44 µg/mL (N. Yu et al., 2017). With respect to depression, another meta-analysis including six studies and 4220 participants found a significant association between depression and lower adiponectin levels, with a difference of −5.00 µg/mL compared to controls, among European participants (Hu et al., 2015).”
4/ Chapter 4 linking Adiponectin, Exercise and Depression (lines 308-313): Although the manuscript is dedicated to Adiponectin, the Authors need to discuss that the beneficial effects of exercise may also be mediated through other mediators than Adiponectin. Adiponectin contributes to a global effect of exercise that decreases several other mediators of stress. It is a question of homeostasis between levels of indicators of depression and anti-depression signals. This point needs to be discussed to better consider the place/roles of Adiponectin among different signals regulated by exercise and related to the control of stress.
Response 4. We thank the reviewer for asking this important clarification. We have added a general discussion describing the main pathways and molecular mediators involved in the positive effects of physical exercise, emphasizing that the mechanism of action of adiponectin is closely interconnected with other compounds and signaling pathways, such as BDNF, irisin and β-endorphins (page 9, line 386-400). The new paragraph reads: “Exercise modulates multiple molecular pathways that operate both at the metabolic and central nervous system levels. It triggers a cascade of signals that collectively regulate the balance between stress- and anti-stress mediators. This includes increased circulating levels of BDNF (Blume & Royes, 2024), irisin (Sadier et al., 2024) and β-endorphins (Pilozzi et al., 2020), together with decreased concentrations of pro-inflammatory cytokines and cortisol (Calcaterra et al., 2022), which contribute to an overall homeostatic shift toward resilience leading to an improved mood regulation. In this context, adiponectin acts synergistically within this complex network by enhancing insulin sensitivity, reducing neuroinflammation and supporting BDNF signalling. Along with other exercise-induced mediators, such as irisin and insulin (Passos & Gonçalves, 2014), adiponectin helps to maintain metabolic balance and promote neuronal health. Together, all these molecules enhance brain energy metabolism, increase neurotrophic support and foster resilience to stress (Nowacka-Chmielewska et al., 2022). Therefore, the antidepressant effects of exercise are more likely to arise from the integrated action of multiple mediators, with adiponectin representing one of the key components linking peripheral metabolic adaptations to central mechanisms of neuroplasticity.”
5/ Chapter 4 linking Adiponectin, Exercise and Depression (Figure 2 and the associated text): It would be relevant to complete the text and the Figure 2 with the description of some molecular targets involved in oxidative stress regulation, given that Adiponectin is also known to exert antioxidant activities. Indeed, oxidative stress and inflammation are known to act bidirectionally (as mentioned Line 40) and amplify cellular dysfunctions during depression. Thus the dual anti-inflammatory and antioxidant properties of Adiponectin may be involved.
Response 5: We thank the reviewer for this interesting suggestion. We have added a point in Figure 2 and in the corresponding text explaining the antioxidant role of adiponectin. In addition, we have included this aspect among the molecular mechanisms related to adiponectin’s effects, as (VI) antioxidative and mitochondrial-protective pathways (page 11, line 473-474). The revised paragraph (page, 12-13, line 548-566) now reads as follows: “Adiponectin exerts potent antioxidant and mitochondrial-protective effects that are fundamental for neuroprotection. Kadowaki and colleagues demonstrated that oxidative stress is markedly elevated in AdipoR1- and AdipoR2-deficient mice, providing compelling evidence that the adiponectin–AdipoRs signalling pathway plays a pivotal role in counteracting oxidative damage and maintaining redox balance (Kadowaki, 2006). Experimental evidence indicates that adiponectin reduces the generation of reactive oxygen species (ROS) primarily by enhancing the activity of key antioxidant enzymes such as superoxide dismutase (SOD), catalase and glutathione peroxidase, thereby strengthening endogenous antioxidant defences (Gradinaru et al., 2017). Beyond its antioxidant functions, adiponectin also regulates mitochondrial homeostasis and quality control, mechanisms essential for sustaining neuronal energy metabolism. Activation of the AdipoR1 - PGC-1α signaling axis has been shown to prevent mitochondrial oxidative damage, restore mitochondrial biogenesis and promote the removal of damaged mitochondria via mitophagy, thus limiting ROS accumulation and preserving mitochondrial efficiency (Iwabu et al., 2010; Wang et al., 2018; Wu et al., 2020; J. Yu et al., 2019). Moreover, the suppression of NF-κB signalling due to the anti-inflammatory effects of adiponectin further reduces ROS generation and prevents neuronal cytotoxicity (Chan et al., 2012). Through these combined actions, adiponectin lowers oxidative stress, preserves mitochondrial function, mitigates cellular injury and contributes to neuronal survival and synaptic plasticity within the hippocampus.”
Minor Revisions:
6/ Lines 9016-917. The reference 128 needs to be formatted to delete “(Noisy-le-grand)” indication.
Response. We thank the reviewer for noticing this. We have corrected the formatting of reference 128.
References
Blume, G. R., & Royes, L. F. F. (2024). Peripheral to brain and hippocampus crosstalk induced by exercise mediates cognitive and structural hippocampal adaptations. Life Sciences, 352, 122799. https://doi.org/10.1016/j.lfs.2024.122799
Bocchio-Chiavetto, L., Bagnardi, V., Zanardini, R., Molteni, R., Gabriela Nielsen, M., Placentino, A., Giovannini, C., Rillosi, L., Ventriglia, M., Riva, M. A., & Gennarelli, M. (2010). Serum and plasma BDNF levels in major depression: A replication study and meta-analyses. The World Journal of Biological Psychiatry, 11(6), 763–773. https://doi.org/10.3109/15622971003611319
Calcaterra, V., Vandoni, M., Rossi, V., Berardo, C., Grazi, R., Cordaro, E., Tranfaglia, V., Carnevale Pellino, V., Cereda, C., & Zuccotti, G. (2022). Use of Physical Activity and Exercise to Reduce Inflammation in Children and Adolescents with Obesity. International Journal of Environmental Research and Public Health, 19(11), 6908. https://doi.org/10.3390/ijerph19116908
Chan, K.-H., Lam, K. S.-L., Cheng, O.-Y., Kwan, J. S.-C., Ho, P. W.-L., Cheng, K. K.-Y., Chung, S. K., Ho, J. W.-M., Guo, V. Y., & Xu, A. (2012). Adiponectin is Protective against Oxidative Stress Induced Cytotoxicity in Amyloid-Beta Neurotoxicity. PLoS ONE, 7(12), e52354. https://doi.org/10.1371/journal.pone.0052354
Correia, A. S., Cardoso, A., & Vale, N. (2023). BDNF Unveiled: Exploring Its Role in Major Depression Disorder Serotonergic Imbalance and Associated Stress Conditions. Pharmaceutics, 15(8), 2081. https://doi.org/10.3390/pharmaceutics15082081
Gradinaru, D., Margina, D., Borsa, C., Ionescu, C., Ilie, M., Costache, M., Dinischiotu, A., & Prada, G.-I. (2017). Adiponectin: Possible link between metabolic stress and oxidative stress in the elderly. Aging Clinical and Experimental Research, 29(4), 621–629. https://doi.org/10.1007/s40520-016-0629-z
Hu, Y., Dong, X., & Chen, J. (2015). Adiponectin and depression: A meta-analysis. Biomedical Reports, 3(1), 38–42. https://doi.org/10.3892/br.2014.372
Iwabu, M., Yamauchi, T., Okada-Iwabu, M., Sato, K., Nakagawa, T., Funata, M., Yamaguchi, M., Namiki, S., Nakayama, R., Tabata, M., Ogata, H., Kubota, N., Takamoto, I., Hayashi, Y. K., Yamauchi, N., Waki, H., Fukayama, M., Nishino, I., Tokuyama, K., … Kadowaki, T. (2010). Adiponectin and AdipoR1 regulate PGC-1alpha and mitochondria by Ca(2+) and AMPK/SIRT1. Nature, 464(7293), 1313–1319. https://doi.org/10.1038/nature08991
Jonas, M. I., Kurylowicz, A., Bartoszewicz, Z., Lisik, W., Jonas, M., Domienik-Karlowicz, J., & Puzianowska-Kuznicka, M. (2017). Adiponectin/resistin interplay in serum and in adipose tissue of obese and normal-weight individuals. Diabetology & Metabolic Syndrome, 9(1), 95. https://doi.org/10.1186/s13098-017-0293-2
Kadowaki, T. (2006). Adiponectin and adiponectin receptors in insulin resistance, diabetes, and the metabolic syndrome. Journal of Clinical Investigation, 116(7), 1784–1792. https://doi.org/10.1172/JCI29126
Kishi, T., Yoshimura, R., Ikuta, T., & Iwata, N. (2018). Brain-Derived Neurotrophic Factor and Major Depressive Disorder: Evidence from Meta-Analyses. Frontiers in Psychiatry, 8, 308. https://doi.org/10.3389/fpsyt.2017.00308
Nowacka-Chmielewska, M., Grabowska, K., Grabowski, M., Meybohm, P., Burek, M., & Małecki, A. (2022). Running from Stress: Neurobiological Mechanisms of Exercise-Induced Stress Resilience. International Journal of Molecular Sciences, 23(21), 13348. https://doi.org/10.3390/ijms232113348
Passos, M., & Gonçalves, M. (2014). Regulation of Insulin Sensitivity by Adiponectin and its Receptors in Response to Physical Exercise. Hormone and Metabolic Research, 46(09), 603–608. https://doi.org/10.1055/s-0034-1377026
Pilozzi, A., Carro, C., & Huang, X. (2020). Roles of β-Endorphin in Stress, Behavior, Neuroinflammation, and Brain Energy Metabolism. International Journal of Molecular Sciences, 22(1), 338. https://doi.org/10.3390/ijms22010338
Sadier, N. S., El Hajjar, F., Al Sabouri, A. A. K., Abou-Abbas, L., Siomava, N., Almutary, A. G., & Tambuwala, M. M. (2024). Irisin: An unveiled bridge between physical exercise and a healthy brain. Life Sciences, 339, 122393. https://doi.org/10.1016/j.lfs.2023.122393
Wang, B., Guo, H., Li, X., Yue, L., Liu, H., Zhao, L., Bai, H., Liu, X., Wu, X., & Qu, Y. (2018). Adiponectin Attenuates Oxygen–Glucose Deprivation-Induced Mitochondrial Oxidative Injury and Apoptosis in Hippocampal HT22 Cells via the JAK2/STAT3 Pathway. Cell Transplantation, 27(12), 1731–1743. https://doi.org/10.1177/0963689718779364
Wolkowitz, O. M., Wolf, J., Shelly, W., Rosser, R., Burke, H. M., Lerner, G. K., Reus, V. I., Nelson, J. C., Epel, E. S., & Mellon, S. H. (2011). Serum BDNF levels before treatment predict SSRI response in depression. Progress in Neuro-Psychopharmacology & Biological Psychiatry, 35(7), 1623–1630. https://doi.org/10.1016/j.pnpbp.2011.06.013
Wu, X., Luo, J., Liu, H., Cui, W., Guo, K., Zhao, L., Bai, H., Guo, W., Guo, H., Feng, D., & Qu, Y. (2020). Recombinant Adiponectin Peptide Ameliorates Brain Injury Following Intracerebral Hemorrhage by Suppressing Astrocyte-Derived Inflammation via the Inhibition of Drp1-Mediated Mitochondrial Fission. Translational Stroke Research, 11(5), 924–939. https://doi.org/10.1007/s12975-019-00768-x
Yu, J., Zheng, J., Lu, J., Sun, Z., Wang, Z., & Zhang, J. (2019). AdipoRon Protects Against Secondary Brain Injury After Intracerebral Hemorrhage via Alleviating Mitochondrial Dysfunction: Possible Involvement of AdipoR1–AMPK–PGC1α Pathway. Neurochemical Research, 44(7), 1678–1689. https://doi.org/10.1007/s11064-019-02794-5
Yu, N., Ruan, Y., Gao, X., & Sun, J. (2017). Systematic Review and Meta-Analysis of Randomized, Controlled Trials on the Effect of Exercise on Serum Leptin and Adiponectin in Overweight and Obese Individuals. Hormone and Metabolic Research, 49(03), 164–173. https://doi.org/10.1055/s-0042-121605
Reviewer 2 Report
Comments and Suggestions for Authors
This paper is deemed to be a nice summary. The following suggestions to potentially improve the paper can be considered:
- Most importantly, the methods of selection of these cited references were not clearly stated. The paper by authority’s opinion is not always placed in the good position during the EBM period.
- This summary report focuses on adiponectin in relation to inflammation. Others may prefer the relation of adiponectin to oxidative stress as adiponectin has antioxidant effects. This might promote more description of the relation of adiponectin to oxidative stress.
- More description of the relation of adiponectin to neurotransmitters (i.e., serotonin, dopamine) would be expressed to be added.
- Some genetic variants of adiponectin have been known. The variants’ effects of adiponectin expression could be added.
- The degree and types of exercise might be detailed from the earlier literature of adiponectin.
- High levels of adiponectin in the blood are reported to worsen cognitive disorders and so on. The paradoxical effects of adiponectin could be added.
- If possible, some comments may be useful; how is the hypothesis about whether or not the anti-inflammatory drugs as NSAIDs are potential medication related to adiponectin?
- If possible, some comments may be useful; how is the hypothesis about whether or not the anti-oxidative supplements as cumin are potential medication related to adiponectin?
- Ref lists; the abbreviated or full spelling names of Journals could be unified (e.g., no. 19, 20 etc. may be abbreviated).
- The terms GLP1RA and GLP-1RA were mixed.
Author Response
This paper is deemed to be a nice summary.
We thank the reviewer for the appreciation of our work.
The following suggestions to potentially improve the paper can be considered:
Comments 1. Most importantly, the methods of selection of these cited references were not clearly stated. The paper by authority’s opinion is not always placed in the good position during the EBM period.
Response 1: We thank the reviewer for this suggestion. As our paper is a narrative rather than a systematic review, we did not follow formal selection protocols such as PRISMA guidelines. However, to enhance the clarity and transparency of our approach, we have added a brief “Methods” section specifying the main databases consulted (PubMed, Scopus, Web of Science) and the keywords used. This new section has been added on page 3, line 89-95, under the “Methods” paragraph.
“Methods:
To identify relevant studies and publications, a literature search was conducted in PubMed (MEDLINE), Google Scholar and Scopus databases, using keywords including neuroplasticity, hippocampal neurogenesis, depression, physical activity, and adiponectin. Publications in English from 2000 to 2025 were considered. Preference was given to peer-reviewed original research articles and meta-analyses addressing molecular and cellular mechanisms linking adiponectin to depression and neuroplasticity.”
Comments 2. This summary report focuses on adiponectin in relation to inflammation. Others may prefer the relation of adiponectin to oxidative stress as adiponectin has antioxidant effects. This might promote more description of the relation of adiponectin to oxidative stress.
Response 2. We thank the reviewer for this comment. We added an important point about the antioxidant effects of adiponectin on page 12-13 (line 548-566).
“Adiponectin exerts potent antioxidant and mitochondrial-protective effects that are fundamental for neuroprotection. Kadowaki and colleagues demonstrated that oxidative stress is markedly elevated in AdipoR1- and AdipoR2-deficient mice, providing compelling evidence that the adiponectin–AdipoRs signalling pathway plays a pivotal role in counteracting oxidative damage and maintaining redox balance (Kadowaki, 2006). Experimental evidence indicates that adiponectin reduces the generation of reactive oxygen species (ROS) primarily by enhancing the activity of key antioxidant enzymes such as superoxide dismutase (SOD), catalase and glutathione peroxidase, thereby strengthening endogenous antioxidant defences (Gradinaru et al., 2017). Beyond its antioxidant functions, adiponectin also regulates mitochondrial homeostasis and quality control, mechanisms essential for sustaining neuronal energy metabolism. Activation of the AdipoR1 - PGC-1α signalling axis has been shown to prevent mitochondrial oxidative damage, restore mitochondrial biogenesis and promote the removal of damaged mitochondria via mitophagy, thus limiting ROS accumulation and preserving mitochondrial efficiency (Iwabu et al., 2010; B. Wang et al., 2018; Wu et al., 2020; Yu et al., 2019). Moreover, the suppression of NF-κB signalling due to the anti-inflammatory effects of adiponectin further reduces ROS generation and prevents neuronal cytotoxicity (Chan et al., 2012). Through these combined actions, adiponectin lowers oxidative stress, preserves mitochondrial function, mitigates cellular injury and contributes to neuronal survival and synaptic plasticity within the hippocampus.”
Comments 3. More description of the relation of adiponectin to neurotransmitters (i.e., serotonin, dopamine) would be expressed to be added.
Response 3. We thank the reviewer for this suggestion. We have expanded our discussion to better explain how adiponectin interacts with serotonergic, dopaminergic and glutamatergic pathways on page 7 (line 303-316). The new paragraph reads: “The expression of AdipoRs in the brain modulates the function of several neurotransmitters and neuropeptides. Findings show that AdipoR1 is present in serotonin neurons in the dorsal raphe nucleus, where deletion of AdipoR1 in these area leads to reduced expression of the serotonin synthesis enzyme Tryptophan Hydroxylase 2 (TPH2), decreased serotonin levels and altered Serotonin Transporter (SERT) expression, resulting in impaired serotonin transmission and depression-like behaviours (Li et al., 2021). Similarly, dopamine neurons in the ventral tegmental area express AdipoR1 and loss of AdipoR1 pathway activity has been shown to increase dopamine activity and anxiety-like behaviours (Bloemer et al., 2019; F. Sun et al., 2019). Regarding glutamatergic transmission, recent evidence shows that adiponectin knockout mice display cognitive deficits and impaired synaptic plasticity in the hippocampus, accompanied by altered levels of presynaptic and postsynaptic proteins involved in glutamatergic neurotransmission. These deficits can be rescued by adiponectin receptor agonists, indicating a regulatory role for adiponectin in the glutamate receptor expression and synaptic function (Bloemer et al., 2019; Z. Sun et al., 2023).”
Comments 4. Some genetic variants of adiponectin have been known. The variants’ effects of adiponectin expression could be added.
Response 4. We thank the reviewer for this valuable suggestion. We have identified evidence supporting the role of specific genetic variants in modulating the response to physical exercise and adiponectin production. Accordingly, we have added a discussion of this aspect, which was not previously considered, in the revised manuscript (page 8-9, line 364-379), as follow: “It is important to note that differences in adiponectin production depend not only on lifestyle-related factors but also on an individual’s genetic background, particularly on genetic variants within the ADIPOQ gene, which can influence adiponectin expression, circulating levels and biological activity. Several studies across diverse populations have investigated the impact of ADIPOQ polymorphisms on variations in adiponectin concentrations, insulin sensitivity, diabetes susceptibility and, importantly, on the physiological response to physical exercise. Notably, multiple ADIPOQ variants, including rs2241766 (Lee et al., 2013), rs1501299 (Cardozo Gasparin et al., 2024; Lee et al., 2013; LeoÅ„ska-Duniec et al., 2018), rs16861205 (Siitonen et al., 2011), rs266729 (LeoÅ„ska-Duniec et al., 2018; Shari et al., 2024) and rs17300539 (Corbi et al., 2019; De Luis Roman et al., 2023) have been associated with longitudinal changes in serum adiponectin levels, BMI, fat mass and lipid profile following exercise interventions. Importantly, specific allele carriers tend to exhibit greater reductions in body fat and more pronounced metabolic improvements after aerobic training compared to others, for example, C allele carriers of rs266729 (Shari et al., 2024) and homozygous A allele carriers of rs17300539 (Corbi et al., 2019; De Luis Roman et al., 2023). These findings emphasize the role of ADIPOQ genetic variability in modulating exercise responsiveness and can partly explain the substantial interindividual variability observed in exercise-induced metabolic outcomes.”
Comments 5. The degree and types of exercise might be detailed from the earlier literature of adiponectin.
Response 5. We thank the reviewer for this helpful suggestion. We have thus expanded this point to provide a more detailed description of the differences in adiponectin secretion following various types of exercise (acute, chronic and endurance) on page 8 (line 347-363), as follow: “Physical exercise modulates adiponectin production in a manner strongly dependent on its type, intensity and duration. Aerobic and endurance training, particularly when performed regularly over 3 months consistently elevates circulating adiponectin levels (Bruun et al., 2006), reflecting improved metabolic and anti-inflammatory adaptations. Acute, high-intensity exercise can trigger transient increases in adiponectin, especially in young and fit individuals (Bouassida et al., 2010; Fonseca et al., 2021; Jürimäe et al., 2006; Kraemer et al., 2003; Mallardo et al., 2024; Nishida et al., 2019), although findings remain inconsistent across studies, with some reporting modest rises and others no significant changes or even slight reductions (Ferguson et al., 2004; Jamurtas et al., 2006; Jürimäe et al., 2006). In contrast, chronic moderate-intensity training produces more stable and pronounced increases in adiponectin, particularly in individuals with obesity or type 2 diabetes (Becic et al., 2018; Blüher et al., 2006; X. Wang et al., 2015). Light-intensity physical activity may also enhance adiponectin levels, although responses tend to be more variable in mixed populations (Nishida et al., 2019). Overall, aerobic modalities appear more effective than combined training in raising adiponectin concentrations, though the magnitude of change often depends on the duration of the intervention, the baseline metabolic status and other individual characteristics (Andarianto et al., 2024; Becic et al., 2018; Putra et al., 2025; Sirico et al., 2018).
Combined training can also lead to increases in adiponectin, particularly when paired with aerobic exercise. However, these effects are generally less consistent and smaller compared to aerobic training alone (Andarianto et al., 2024; Bouassida et al., 2010; Mallardo et al., 2023; Simpson & Singh, 2008).
Comments 6. High levels of adiponectin in the blood are reported to worsen cognitive disorders and so on. The paradoxical effects of adiponectin could be added.
Response 6. We thank the reviewer for this suggestion. Although this paradoxical effect has been mainly observed in neurodegenerative disorders rather than in depression, we have added a few lines about this effect on page 7-8 (line 317-327): “Although adiponectin is generally referred to an anti-inflammatory and neuroprotective molecule, several clinical studies have described the so-called “adiponectin paradox”, whereby higher circulating adiponectin levels are paradoxically associated with cognitive decline, Alzheimer’s disease and frailty in older adults. In more details, different studies have reported that elevated adiponectin levels predict incident Alzheimer’s disease, especially in women and in individuals with amyloid pathology or low body mass index (BMI) (Kim et al., 2022; van Himbergen et al., 2012; Wennberg et al., 2016). However, other studies have found that lower adiponectin levels are associated with worse cognitive performance, suggesting a possible protective role of adiponectin in brain functions (Liu et al., 2022; Teixeira et al., 2013). This opposite association highlights the complexity of adiponectin signaling in neurodegenerative processes and underscores the need for further studies.”
Comment 7. If possible, some comments may be useful; how is the hypothesis about whether or not the anti-inflammatory drugs as NSAIDs are potential medication related to adiponectin?
Response 7. We thank the reviewer for addressing this point. To the best of our knowledge, NSAIDs (nonsteroidal anti-inflammatory drugs) are not mentioned in the current literature as medications that act via the adiponectin receptor or its primary signalling pathways (AMPK, PPARα), or as agents that modulate adiponectin levels for therapeutic purposes. However, NSAIDs, by inhibiting cyclooxygenase (COX) enzymes, by reducing prostaglandin synthesis and by downregulating the inflammatory signalling, may indirectly influence adiponectin functions through their anti-inflammatory effects (Na et al., 2023). On the other hand, adiponectin itself reduces inflammatory markers (TNF-α, IL-6) and increases anti-inflammatory cytokines (IL-10) (Jung & Jung, 2021). Therefore, while both NSAIDs and adiponectin exert anti-inflammatory effects, they act through distinct pathways. Indirect interactions are possible, but no direct mechanistic link has been established in the literature, so far. For this reason, we believe that a detailed discussion of NSAIDs is beyond the scope of our review, which primarily focuses on the effects of adiponectin in modulating neuroplasticity.
Comment 8. If possible, some comments may be useful; how is the hypothesis about whether or not the anti-oxidative supplements as cumin are potential medication related to adiponectin?
Response 8. We thank the reviewer for pointing this out. Anti-oxidative supplements, as Cumin (Cuminum cyminum) has been shown to increase adiponectin serum levels in diabetes type 2 patients in one study (Jafari et al., 2017). Similar adiponectin-enhancing effects have been described for other antioxidant nutraceuticals, including curcumin and resveratrol (Janiszewska et al., 2021). Nevertheless, although this is an interesting and valuable point, the current available evidence is limited and not sufficiently related to the main topic of our review, therefore, we decided to not include it in our manuscript.
Comment 9. Ref lists; the abbreviated or full spelling names of Journals could be unified (e.g., no. 19, 20 etc. may be abbreviated).
Response 9. We thank the reviewer for noticing this. We have corrected and unified all journal names in the reference list.
Comment 10. The terms GLP1RA and GLP-1RA were mixed.
Response 10. We thank the reviewer for the observation. As we have removed this section from our review following another reviewer’s suggestion, the terms previously mentioned are no longer present in the manuscript, and therefore there is no longer any inconsistency in terminology.
References
Andarianto, A., Rejeki, P. S., Pranoto, A., Izzatunnisa, N., Rahmanto, I., Muhammad, M., & Halim, S. (2024). Effects of moderate-intensity combination exercise on increase adiponectin levels, muscle mass, and decrease fat mass in obese women. Retos, 55, 296–301. https://doi.org/10.47197/retos.v55.103738
Becic, T., Studenik, C., & Hoffmann, G. (2018). Exercise Increases Adiponectin and Reduces Leptin Levels in Prediabetic and Diabetic Individuals: Systematic Review and Meta-Analysis of Randomized Controlled Trials. Medical Sciences, 6(4), 97. https://doi.org/10.3390/medsci6040097
Bloemer, J., Pinky, P. D., Smith, W. D., Bhattacharya, D., Chauhan, A., Govindarajulu, M., Hong, H., Dhanasekaran, M., Judd, R., Amin, R. H., Reed, M. N., & Suppiramaniam, V. (2019). Adiponectin Knockout Mice Display Cognitive and Synaptic Deficits. Frontiers in Endocrinology, 10, 819. https://doi.org/10.3389/fendo.2019.00819
Blüher, M., Bullen, J. W., Lee, J. H., Kralisch, S., Fasshauer, M., Klöting, N., Niebauer, J., Schön, M. R., Williams, C. J., & Mantzoros, C. S. (2006). Circulating Adiponectin and Expression of Adiponectin Receptors in Human Skeletal Muscle: Associations with Metabolic Parameters and Insulin Resistance and Regulation by Physical Training. The Journal of Clinical Endocrinology & Metabolism, 91(6), 2310–2316. https://doi.org/10.1210/jc.2005-2556
Bouassida, A., Chamari, K., Zaouali, M., Feki, Y., Zbidi, A., & Tabka, Z. (2010). Review on leptin and adiponectin responses and adaptations to acute and chronic exercise. British Journal of Sports Medicine, 44(9), 620–630. https://doi.org/10.1136/bjsm.2008.046151
Bruun, J. M., Helge, J. W., Richelsen, B., & Stallknecht, B. (2006). Diet and exercise reduce low-grade inflammation and macrophage infiltration in adipose tissue but not in skeletal muscle in severely obese subjects. American Journal of Physiology. Endocrinology and Metabolism, 290(5), E961-967. https://doi.org/10.1152/ajpendo.00506.2005
Cardozo Gasparin, C., Leite, N., Lehtonen Rodrigues De Souza, R., Viater Tureck, L., E. Milano-Gai, G., Pizzi, J., R. Silva, L., De Fátima Aguiar Lopes, M., A. Lopes, W., & Furtado-Alle, L. (2024). A relationship between Single Nucleotide Polymorphism (SNP) in HSD11β1 and ADIPOQ genes and obesity related features in children and adolescents submitted on physical exercises. Brazilian Journal of Implantology and Health Sciences, 6(1), 1791–1810. https://doi.org/10.36557/2674-8169.2024v6n1p1791-1810
Chan, K.-H., Lam, K. S.-L., Cheng, O.-Y., Kwan, J. S.-C., Ho, P. W.-L., Cheng, K. K.-Y., Chung, S. K., Ho, J. W.-M., Guo, V. Y., & Xu, A. (2012). Adiponectin is Protective against Oxidative Stress Induced Cytotoxicity in Amyloid-Beta Neurotoxicity. PLoS ONE, 7(12), e52354. https://doi.org/10.1371/journal.pone.0052354
Corbi, G., Polito, R., Monaco, M. L., Cacciatore, F., Scioli, M., Ferrara, N., Daniele, A., & Nigro, E. (2019). Adiponectin Expression and Genotypes in Italian People with Severe Obesity Undergone a Hypocaloric Diet and Physical Exercise Program. Nutrients, 11(9), 2195. https://doi.org/10.3390/nu11092195
De Luis Roman, D., Izaola Jauregui, O., & Primo Martin, D. (2023). The Polymorphism rs17300539 in the Adiponectin Promoter Gene Is Related to Metabolic Syndrome, Insulin Resistance, and Adiponectin Levels in Caucasian Patients with Obesity. Nutrients, 15(24), 5028. https://doi.org/10.3390/nu15245028
Ferguson, M. A., White, L. J., McCoy, S., Kim, H.-W., Petty, T., & Wilsey, J. (2004). Plasma adiponectin response to acute exercise in healthy subjects. European Journal of Applied Physiology, 91(2–3), 324–329. https://doi.org/10.1007/s00421-003-0985-1
Fonseca, T. R., Mendes, T. T., Ramos, G. P., Cabido, C. E. T., Morandi, R. F., Ferraz, F. O., Miranda, A. S., Mendonça, V. A., Teixeira, A. L., Silami-Garcia, E., Nunes-Silva, A., & Teixeira, M. M. (2021). Aerobic Training Modulates the Increase in Plasma Concentrations of Cytokines in response to a Session of Exercise. Journal of Environmental and Public Health, 2021, 1304139. https://doi.org/10.1155/2021/1304139
Gradinaru, D., Margina, D., Borsa, C., Ionescu, C., Ilie, M., Costache, M., Dinischiotu, A., & Prada, G.-I. (2017). Adiponectin: Possible link between metabolic stress and oxidative stress in the elderly. Aging Clinical and Experimental Research, 29(4), 621–629. https://doi.org/10.1007/s40520-016-0629-z
Iwabu, M., Yamauchi, T., Okada-Iwabu, M., Sato, K., Nakagawa, T., Funata, M., Yamaguchi, M., Namiki, S., Nakayama, R., Tabata, M., Ogata, H., Kubota, N., Takamoto, I., Hayashi, Y. K., Yamauchi, N., Waki, H., Fukayama, M., Nishino, I., Tokuyama, K., … Kadowaki, T. (2010). Adiponectin and AdipoR1 regulate PGC-1alpha and mitochondria by Ca(2+) and AMPK/SIRT1. Nature, 464(7293), 1313–1319. https://doi.org/10.1038/nature08991
Jafari, S., Sattari, R., & Ghavamzadeh, S. (2017). Evaluation the effect of 50 and 100 mg doses of Cuminum cyminum essential oil on glycemic indices, insulin resistance and serum inflammatory factors on patients with diabetes type II: A double-blind randomized placebo-controlled clinical trial. Journal of Traditional and Complementary Medicine, 7(3), 332–338. https://doi.org/10.1016/j.jtcme.2016.08.004
Jamurtas, A. Z., Theocharis, V., Koukoulis, G., Stakias, N., Fatouros, I. G., Kouretas, D., & Koutedakis, Y. (2006). The effects of acute exercise on serum adiponectin and resistin levels and their relation to insulin sensitivity in overweight males. European Journal of Applied Physiology, 97(1), 122–126. https://doi.org/10.1007/s00421-006-0169-x
Janiszewska, J., Ostrowska, J., & Szostak-WÄ™gierek, D. (2021). The Influence of Nutrition on Adiponectin—A Narrative Review. Nutrients, 13(5), 1394. https://doi.org/10.3390/nu13051394
Jung, H. N., & Jung, C. H. (2021). The Role of Anti-Inflammatory Adipokines in Cardiometabolic Disorders: Moving beyond Adiponectin. International Journal of Molecular Sciences, 22(24), 13529. https://doi.org/10.3390/ijms222413529
Jürimäe, J., Purge, P., & Jürimäe, T. (2006). Adiponectin and stress hormone responses to maximal sculling after volume-extended training season in elite rowers. Metabolism: Clinical and Experimental, 55(1), 13–19. https://doi.org/10.1016/j.metabol.2005.06.020
Kadowaki, T. (2006). Adiponectin and adiponectin receptors in insulin resistance, diabetes, and the metabolic syndrome. Journal of Clinical Investigation, 116(7), 1784–1792. https://doi.org/10.1172/JCI29126
Kim, K. Y., Ha, J., Kim, M., Cho, S. Y., Kim, H., Kim, E., & for the Alzheimer’s Disease Neuroimaging Initiative. (2022). Plasma adiponectin levels predict cognitive decline and cortical thinning in mild cognitive impairment with beta-amyloid pathology. Alzheimer’s Research & Therapy, 14(1), 165. https://doi.org/10.1186/s13195-022-01107-3
Kraemer, R. R., Aboudehen, K. S., Carruth, A. K., Durand, R. T. J., Acevedo, E. O., Hebert, E. P., Johnson, L. G., & Castracane, V. D. (2003). Adiponectin responses to continuous and progressively intense intermittent exercise. Medicine and Science in Sports and Exercise, 35(8), 1320–1325. https://doi.org/10.1249/01.MSS.0000079072.23998.F3
Lee, K.-Y., Kang, H.-S., & Shin, Y.-A. (2013). Exercise improves adiponectin concentrations irrespective of the adiponectin gene polymorphisms SNP45 and the SNP276 in obese Korean women. Gene, 516(2), 271–276. https://doi.org/10.1016/j.gene.2012.12.028
LeoÅ„ska-Duniec, A., Grzywacz, A., JastrzÄ™bski, Z., Jażdżewska, A., LuliÅ„ska-Kuklik, E., Moska, W., Leźnicka, K., Ficek, K., Rzeszutko, A., Dornowski, M., & CiÄ™szczyk, P. (2018). ADIPOQ polymorphisms are associated with changes in obesityrelatedtraits in response to aerobic training programme in women. Biology of Sport, 35(2), 165–173. https://doi.org/10.5114/biolsport.2018.72762
Li, C., Meng, F., Garza, J. C., Liu, J., Lei, Y., Kirov, S. A., Guo, M., & Lu, X.-Y. (2021). Modulation of depression-related behaviors by adiponectin AdipoR1 receptors in 5-HT neurons. Molecular Psychiatry, 26(8), 4205–4220. https://doi.org/10.1038/s41380-020-0649-0
Liu, F., Xu, H., Yin, Y., Chen, Y., Xie, L., Li, H., Wang, D., & Shi, B. (2022). Decreased Adiponectin Levels Are a Risk Factor for Cognitive Decline in Spinal Cord Injury. Disease Markers, 2022, 1–6. https://doi.org/10.1155/2022/5389162
Mallardo, M., D’Alleva, M., Lazzer, S., Giovanelli, N., Graniero, F., Billat, V., Fiori, F., Marinoni, M., Parpinel, M., Daniele, A., & Nigro, E. (2023). Improvement of adiponectin in relation to physical performance and body composition in young obese males subjected to twenty-four weeks of training programs. Heliyon, 9(5), e15790. https://doi.org/10.1016/j.heliyon.2023.e15790
Mallardo, M., Tommasini, E., Missaglia, S., Pecci, C., Rampinini, E., Bosio, A., Morelli, A., Daniele, A., Nigro, E., & Tavian, D. (2024). Effects of Exhaustive Exercise on Adiponectin and High-Molecular-Weight Oligomer Levels in Male Amateur Athletes. Biomedicines, 12(8), 1743. https://doi.org/10.3390/biomedicines12081743
Na, H., Song, Y., & Lee, H.-W. (2023). Emphasis on Adipocyte Transformation: Anti-Inflammatory Agents to Prevent the Development of Cancer-Associated Adipocytes. Cancers, 15(2), 502. https://doi.org/10.3390/cancers15020502
Nishida, Y., Higaki, Y., Taguchi, N., Hara, M., Nakamura, K., Nanri, H., Imaizumi, T., Sakamoto, T., Shimanoe, C., Horita, M., Shinchi, K., & Tanaka, K. (2019). Intensity-Specific and Modified Effects of Physical Activity on Serum Adiponectin in a Middle-Aged Population. Journal of the Endocrine Society, 3(1), 13–26. https://doi.org/10.1210/js.2018-00255
Putra, D. P., Wibawa, J. C., Rossa, M., & Riyono, A. (2025). Effect of physical exercise on adiponectin levels in humans: A systematic review. Fizjoterapia Polska, 25(2), 436–441. https://doi.org/10.56984/8ZG00E1IVU4
Shari, M., Md. Yusof, S., Raja Hussain, R. N. J., Kek, T. L., Mohd Idris N, N., Aiman, S., Md Radzi, N. A. A., Abu Kasim, N. A., Mohamed, M. N., Mazaulan, M., & Syed Mud Puad, S. M. (2024). Aqua Exercises and Adipoq Gene Polymorphism: Impacts on The Metabolic and Obesity-Related Traits Among Obese Women. International Journal of Academic Research in Business and Social Sciences, 14(1), Pages 390-401. https://doi.org/10.6007/IJARBSS/v14-i1/19435
Siitonen, N., Pulkkinen, L., Lindström, J., Kolehmainen, M., Eriksson, J. G., Venojärvi, M., Ilanne-Parikka, P., Keinänen-Kiukaanniemi, S., Tuomilehto, J., & Uusitupa, M. (2011). Association of ADIPOQ gene variants with body weight, type 2 diabetes and serum adiponectin concentrations: The Finnish Diabetes Prevention Study. BMC Medical Genetics, 12(1), 5. https://doi.org/10.1186/1471-2350-12-5
Simpson, K. A., & Singh, M. A. F. (2008). Effects of Exercise on Adiponectin: A Systematic Review. Obesity, 16(2), 241–256. https://doi.org/10.1038/oby.2007.53
Sirico, F., Bianco, A., D’Alicandro, G., Castaldo, C., Montagnani, S., Spera, R., Di Meglio, F., & Nurzynska, D. (2018). Effects of Physical Exercise on Adiponectin, Leptin, and Inflammatory Markers in Childhood Obesity: Systematic Review and Meta-Analysis. Childhood Obesity (Print), 14(4), 207–217. https://doi.org/10.1089/chi.2017.0269
Sun, F., Lei, Y., You, J., Li, C., Sun, L., Garza, J., Zhang, D., Guo, M., Scherer, P. E., Lodge, D., & Lu, X.-Y. (2019). Adiponectin modulates ventral tegmental area dopamine neuron activity and anxiety-related behavior through AdipoR1. Molecular Psychiatry, 24(1), 126–144. https://doi.org/10.1038/s41380-018-0102-9
Sun, Z., Wang, M., Xu, L., Li, Q., Zhao, Z., Liu, X., Meng, F., Liu, J., Wang, W., Li, C., & Jiang, S. (2023). PPARγ/Adiponectin axis attenuates methamphetamine-induced conditional place preference via the hippocampal AdipoR1 signaling pathway. Progress in Neuro-Psychopharmacology and Biological Psychiatry, 125, 110758. https://doi.org/10.1016/j.pnpbp.2023.110758
Teixeira, A. L., Diniz, B. S., Campos, A. C., Miranda, A. S., Rocha, N. P., Talib, L. L., Gattaz, W. F., & Forlenza, O. V. (2013). Decreased Levels of Circulating Adiponectin in Mild Cognitive Impairment and Alzheimer’s Disease. NeuroMolecular Medicine, 15(1), 115–121. https://doi.org/10.1007/s12017-012-8201-2
van Himbergen, T. M., Beiser, A. S., Ai, M., Seshadri, S., Otokozawa, S., Au, R., Thongtang, N., Wolf, P. A., & Schaefer, E. J. (2012). Biomarkers for insulin resistance and inflammation and the risk for all-cause dementia and alzheimer disease: Results from the Framingham Heart Study. Archives of Neurology, 69(5), 594–600. https://doi.org/10.1001/archneurol.2011.670
Wang, B., Guo, H., Li, X., Yue, L., Liu, H., Zhao, L., Bai, H., Liu, X., Wu, X., & Qu, Y. (2018). Adiponectin Attenuates Oxygen–Glucose Deprivation-Induced Mitochondrial Oxidative Injury and Apoptosis in Hippocampal HT22 Cells via the JAK2/STAT3 Pathway. Cell Transplantation, 27(12), 1731–1743. https://doi.org/10.1177/0963689718779364
Wang, X., You, T., Murphy, K., Lyles, M. F., & Nicklas, B. J. (2015). Addition of Exercise Increases Plasma Adiponectin and Release from Adipose Tissue. Medicine & Science in Sports & Exercise, 47(11), 2450–2455. https://doi.org/10.1249/MSS.0000000000000670
Wennberg, A. M. V., Gustafson, D., Hagen, C. E., Roberts, R. O., Knopman, D., Jack, C., Petersen, R. C., & Mielke, M. M. (2016). Serum Adiponectin Levels, Neuroimaging, and Cognition in the Mayo Clinic Study of Aging. Journal of Alzheimer’s Disease, 53(2), 573–581. https://doi.org/10.3233/JAD-151201
Wu, X., Luo, J., Liu, H., Cui, W., Guo, K., Zhao, L., Bai, H., Guo, W., Guo, H., Feng, D., & Qu, Y. (2020). Recombinant Adiponectin Peptide Ameliorates Brain Injury Following Intracerebral Hemorrhage by Suppressing Astrocyte-Derived Inflammation via the Inhibition of Drp1-Mediated Mitochondrial Fission. Translational Stroke Research, 11(5), 924–939. https://doi.org/10.1007/s12975-019-00768-x
Yu, J., Zheng, J., Lu, J., Sun, Z., Wang, Z., & Zhang, J. (2019). AdipoRon Protects Against Secondary Brain Injury After Intracerebral Hemorrhage via Alleviating Mitochondrial Dysfunction: Possible Involvement of AdipoR1–AMPK–PGC1α Pathway. Neurochemical Research, 44(7), 1678–1689. https://doi.org/10.1007/s11064-019-02794-5
Reviewer 3 Report
Comments and Suggestions for Authors
This manuscript provides an in-depth analysis of the existing scientific evidence regarding adiponectin as a mediator of neuroplasticity and its relationship with exercise and depression. The authors have done an excellent job compiling the available literature and presenting it in a clear, well-structured, and easy-to-follow manner.
The only possible weakness identified is that, at the very end, the authors mention that, in addition to adiponectin, GLP1Ra may also have a potential role. This appears somewhat awkward, as the entire review focuses on a single biomarker, with another one introduced only at the conclusion. My suggestion would be to remove this paragraph. However, if the authors consider it necessary to keep, they should revise it to include discussion of other potential biomarkers, not just GLP1Ra.
Author Response
This manuscript provides an in-depth analysis of the existing scientific evidence regarding adiponectin as a mediator of neuroplasticity and its relationship with exercise and depression. The authors have done an excellent job compiling the available literature and presenting it in a clear, well-structured, and easy-to-follow manner.
We sincerely thank the reviewer for the positive feedback and appreciation of our work.
The only possible weakness identified is that, at the very end, the authors mention that, in addition to adiponectin, GLP1Ra may also have a potential role. This appears somewhat awkward, as the entire review focuses on a single biomarker, with another one introduced only at the conclusion. My suggestion would be to remove this paragraph. However, if the authors consider it necessary to keep, they should revise it to include discussion of other potential biomarkers, not just GLP1Ra.
Response. We thank the reviewer for noticing this point. We agree with the comment and have modified the last part of our discussion, removing the detailed section on GLP1Ra (page 14).
Round 2
Reviewer 1 Report
Comments and Suggestions for Authors
The proposed revisions are in line with the comments of the Reviewer.
Reviewer 2 Report
Comments and Suggestions for Authors
The paper was much improved.